# Association between PM$_{2.5}$ Exposure and Cardiovascular and Respiratory Hospital Admissions Using Spatial GIS Analysis

Hana Tomášková [1,2], Hana Šlachtová [1,2,*], Andrea Dalecká [1,2,3], Pavla Polaufová [1], Jiří Michalík [1], Ivan Tomášek [1] and Anna Šplíchalová [1]

[1] Department of Occupational Medicine, Institute of Public Health, 702 00 Ostrava, Czech Republic
[2] Department of Epidemiology and Public Health, Faculty of Medicine, University of Ostrava, 701 03 Ostrava, Czech Republic
[3] Research Centre for Toxic Compounds in the Environment, Faculty of Science, Masaryk University, 611 37 Brno, Czech Republic
[*] Correspondence: hana.slachtova@osu.cz; Tel.: +420-553-46-1788

**Abstract:** Particulate Matter (PM) air pollution is a serious concern in the northern Moravia region of the Czech Republic. This study aimed to evaluate the association between the risk of acute hospital admissions for cardiovascular (CVD) and respiratory diseases and PM$_{2.5}$ concentrations using a geographic information system (GIS). The data on acute hospital admissions for cardiovascular (I00-99 according to ICD-10) and respiratory (J00-99) diseases was assigned to 77 geographical units (population of 601,299) based on the residence. The annual concentrations of PM$_{2.5}$ in the period from 2013–2019 were assigned to these units according to the respective concentration iso-shapes. The Incidence Rate Ratio (IRR) and 95% confidence interval (CI) were calculated for each concentration category and then compared with the reference category. Statistical analyses were performed using SW STATA v.15. In 2013, approx. half of the population (56%) belonged to the PM$_{2.5}$ category 34–35 µg·m$^{-3}$, and 4% lived in PM$_{2.5}$ concentrations $\geq$ 38 µg·m$^{-3}$. During the analysed period, the average concentrations decreased from 30.8 to 21.4 µg·m$^{-3}$. A statistically significant risk of acute hospitalization for CVD causes was identified in categories $\geq$ 36 µg·m$^{-3}$, and for respiratory causes from 34–35 µg·m$^{-3}$. With increasing concentrations, the risk of both acute cardiovascular and respiratory hospitalizations increased.

**Keywords:** PM$_{2.5}$ spatial model; cardiovascular and respiratory hospital admissions; GIS analysis; Incidence Rate Ratio; iso-concentration shapes





## 1. Introduction

According to the conclusions of the World Health Organization in 2019, air pollution, together with tobacco smoking, an unhealthy diet and harmful use of alcohol, are the leading causes of death from non-communicable diseases (NCD) [1]. Outdoor and indoor air pollution causes 7 million premature deaths globally each year, including more than 5 million due to NCD [2].

Despite the substantial decrease in air pollutant emissions from industrial sites, concentrations of particulate matter (PM) continued to exceed the EU daily limit values in large parts of Europe [3]. In the European Region, ambient air pollution contributed to nearly 500,000 deaths, including ischemic heart diseases, stroke, chronic obstructive pulmonary disease (COPD) and lung cancer [2,4].

Cumulative evidence from multi- and single-city studies suggests a causal associations between short-term and long-term exposure to PM and several specific health outcomes. Systematic reviews examining the effects of air pollution concluded that particulate matters of less than or equal to 10 µm in aerodynamic diameter (PM$_{10}$) and less than or equal to 2.5 µm in aerodynamic diameter (PM$_{2.5}$) were associated with increased all-cause mortality and specific mortality and morbidity for cardiovascular, cerebrovascular and respiratory

causes [5–8]. According to the literature, exacerbation of the ischemic heart disease, heart failure and myocardial infarction [9,10], stroke [6], pneumonia [11] or COPD [12] are among the most pronounced causes of specific mortality.

It should be pointed out that the evidence for these adverse health effects tends to substantially vary between different regions of the world [6]. The Moravian-Silesian region is known as one of the most polluted areas in Europe, particularly due to its long history of coal mining and a high density of industrial pollution sources. This area is also characterized by a high number of settlements with individual heating using solid fuels and a dense transport infrastructure. Additionally, unfavorable meteorological conditions contribute to the transboundary transport of pollutants across Czech-Polish state borders [13,14]. Due to these conditions, air quality has not complied with the national and WHO standards (WHO global air quality guidelines) over the long term [15].

The previously published study from the Moravian-Silesian region, more precisely from Ostrava, was based on individual health data; however, the relevant $PM_{10}$ exposure estimations were available only from four measurement sites [16]. Another time-series analysis with the specifications of lags in acute PM effects on hospital admissions and mortality was published in the Czech language. This analysis was based on modelled concentrations originating from eight monitoring stations with the available PM data [17].

In the present study, the resolution was refined for 7 years (2013–2019) from more than 600,000 inhabitants living in 77 geographical units. This study analyzes and investigates the association between the modelled values of annual exposures to $PM_{2.5}$ and acute cardiovascular and respiratory hospital admissions.

## 2. Materials and Methods

The time-series study was conducted between January 2013 and December 2019 in the Ostrava (area of 331 $km^2$) and Karvina (area of 356 $km^2$) regions, Czech Republic. The ICD-10 registry records of the daily acute hospital admissions for circulatory system (I00-99) and respiratory (J00-99) diseases were collected from all 7 hospitals in the study areas. The hospital admission was assigned according to the place of residence classified into 77 geographical units in the studied regions (601,299 inhabitants). The number of cases was normalized to 100,000 based on the number of inhabitants living in the cadastral units (CU; source—Czech Statistical Office).

The background for the GIS (Geographical Information System) analysis was based on the Czech Hydrometeorological Institute shape-file data of the annual concentrations of $PM_{2.5}$ in the period from 2013–2019 [18]. The main source of data is the ISKO (Information system of the air quality) relation database of the measured data on air pollution concentrations and chemical composition of atmospheric precipitation. The distribution of the ISKO measurement stations across the study area are presented in Figure 1. The maps are constructed in spatial resolution of $1 \times 1$ km, in a Gauss-Krüger projection. These data were assigned to the geographical units according to the respective concentrations of iso-concentration shapes (step of 2 $\mu g \cdot m^{-3}$, concentrations $\leq 29$ to $\geq 38$ $\mu g \cdot m^{-3}$).

The $PM_{2.5}$ concentrations are presented as means, and the cadastral units and residents as absolute frequencies and percentages. The frequencies of hospital admissions across the geographical units are presented graphically on the maps.

Six concentration categories were created ($\leq 29$ $\mu g \cdot m^{-3}$; 30–31 $\mu g \cdot m^{-3}$; 32–33 $\mu g \cdot m^{-3}$; 34–35 $\mu g \cdot m^{-3}$; 36–37 $\mu g \cdot m^{-3}$; $\geq 38$ $\mu g \cdot m^{-3}$) for this analysis.

The Incidence Rate Ratio (IRR) and 95% confidence interval (CI) were calculated for each concentration category. The incidence in the first category with the lowest $PM_{2.5}$ concentrations ($\leq 29$ $\mu g \cdot m^{-3}$) was used as the reference category. Statistical analyses were performed using the SW STATA v.15.

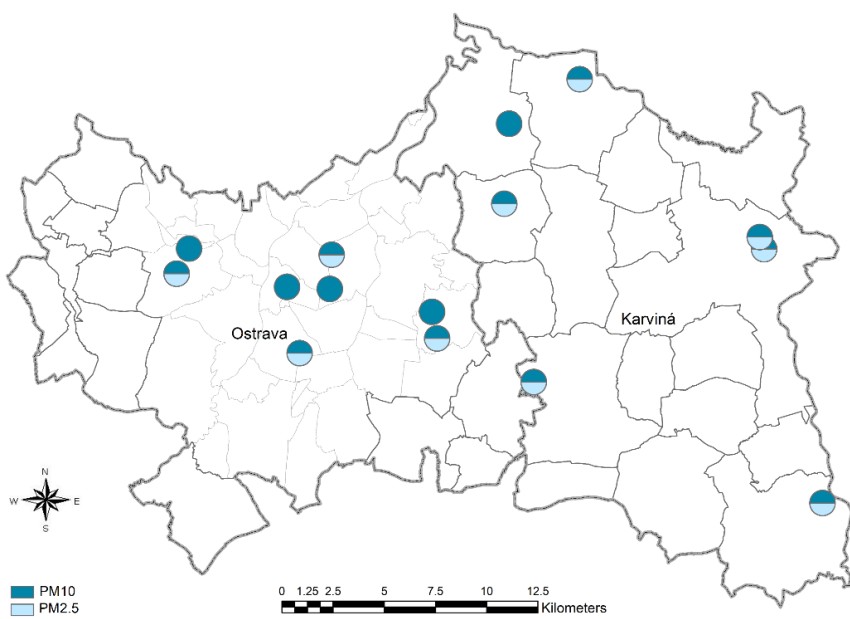

**Figure 1.** The distribution of the ISKO measurement stations across the study areas.

## 3. Results

The annual absolute numbers of hospital admissions in the studied area ranged from 4584 to 5672 for cardiovascular diseases and from 2612 to 3141 for respiratory diseases, respectively. The population decreased from 586,964 to 569,160 during the reporting period. From the total number of 56,959 cases of hospital admissions, 64% were for cardiovascular causes. The proportion of hospitalizations with respiratory diseases was significantly (chi-squared test, $p < 0.001$) higher in women compared to men (38% vs. 33%). Furthermore, people hospitalized for cardiovascular causes were significantly (Mann–Whitney test, $p < 0.001$) older than people hospitalized for respiratory causes (73 years vs. 57 years, respectively).

In 2013, about a half of the population (56%) lived in areas with $PM_{2.5}$ levels of 34–35 $\mu g \cdot m^{-3}$, and 26,000 inhabitants (4%) lived in the areas with $PM_{2.5}$ concentrations $\geq 38$ $\mu g \cdot m^{-3}$ (Table 1). During the analysed period, the average concentration values decreased from 30.8 to 21.4 $\mu g \cdot m^{-3}$ (Table 1).

**Table 1.** Cadastral units in the Ostrava and Karvina districts classified into concentration intervals according to $PM_{2.5}$ (development of the mean concentrations during the monitored years since 2013).

| | CU | Residents | | Mean Concentrations of $PM_{2.5}$ ($\mu g \cdot m^{-3}$) | | | | | |
|---|---|---|---|---|---|---|---|---|---|
| $PM_{2.5}^+$ ($\mu g \cdot m^{-3}$) | n | n | % | 2014 | 2015 | 2016 | 2017 | 2018 | 2019 |
| $\leq 29$ | 4 | 7762 | 1.3% | 26.5 | 23.0 | 22.0 | 22.5 | 24.0 | 20.0 |
| 30–31 | 8 | 56,099 | 9.3% | 28.0 | 25.0 | 24.5 | 25.3 | 26.5 | 20.0 |
| 32–33 | 15 | 74,391 | 12.4% | 29.3 | 26.3 | 24.5 | 26.0 | 27.5 | 20.0 |
| 34–35 | 34 | 335,447 | 55.8% | 31.5 | 28.2 | 26.9 | 28.4 | 31.1 | 21.6 |
| 36–37 | 11 | 101,485 | 16.9% | 33.5 | 30.0 | 29.1 | 30.0 | 31.6 | 22.9 |
| $\geq 38$ | 5 | 26,115 | 4.3% | 32.4 | 29.2 | 29.2 | 30.0 | 32.0 | 23.6 |
| **Celkem** | **77** | **601,299** | **100.0%** | 30.8 | 27.5 | 26.4 | 27.6 | 29.7 | 21.4 |

CU—cadastral units, + assignment of the CU into the concentration interval based on the 2013 year data, n—number.

A statistically significant risk of acute hospitalization for cardiovascular diseases was detected in the categories with $\geq 36$ $\mu g \cdot m^{-3}$ $PM_{2.5}$. The risks of cardiovascular incidence in the areas with the highest concentrations of $PM_{2.5}$ were in most years more than two times higher than in the reference category (Table 2).

**Table 2.** Risk of acute hospital admissions for cardiovascular and respiratory diseases according to the concentration intervals of PM$_{2.5}$.

| | Year | PM$_{2.5}$+ (µg·m$^{-3}$) | <30 | 30–31 | 32–33 | 34–35 | 36–37 | ≥38 |
|---|---|---|---|---|---|---|---|---|
| **Risk of hospitalization from cardiovascular causes (I00–I99)** | 2013 | cases/100,000 | 751.3 | 752.0 | 695.4 | 823.5 | 1321.5 | 2046.6 |
| | | IRR | ref. | **1.01** | **0.93** | **1.10** | **1.77 *** | **2.74 *** |
| | | 95% CI | | (0.76–1.35) | (0.71–1.24) | (0.85–1.46) | (1.36–2.34) | (2.08–3.65) |
| | 2014 | cases/100,000 | 871.5 | 663.8 | 728.2 | 785.1 | 1280.6 | 1881.0 |
| | | IRR | ref. | **0.76** | **0.83** | **0.90** | **1.46*** | **2.15 *** |
| | | 95% CI | | (0.58–1.00) | (0.65–1.09) | (0.70–1.16) | (1.15–1.89) | (1.66–2.81) |
| | 2015 | cases/100,000 | 890.3 | 676.1 | 614.0 | 773.7 | 1124.9 | 2003.0 |
| | | IRR | ref. | **0.76** | **0.69** | **0.87** | **1.27** | **2.25 *** |
| | | 95% CI | | (0.59–1.00) | (0.54–0.90) | (0.69–1.12) | (0.99–1.64) | (1.75–2.94) |
| | 2016 | cases/100,000 | 659.3 | 588.6 | 687.9 | 722.1 | 1146.6 | 1683.8 |
| | | IRR | ref. | **0.90** | **1.05** | **1.10** | **1.75 *** | **2.56 *** |
| | | 95% CI | | (0.67–1.23) | (0.78–1.43) | (0.83–1.48) | (1.32–2.36) | (1.92–3.50) |
| | 2017 | cases/100,000 | 845.3 | 662.5 | 685.6 | 700.4 | 973.5 | 1336.2 |
| | | IRR | ref. | **0.78** | **0.81** | **0.82** | **1.14** | **1.57 *** |
| | | 95% CI | | (0.60–1.03) | (0.62–1.06) | (0.65–1.07) | (0.89–1.49) | (1.21–2.08) |
| | 2018 | cases/100,000 | 813.5 | 567.1 | 664.5 | 669.4 | 815.1 | 1441.2 |
| | | IRR | ref. | **0.70** | **0.78** | **0.79** | **0.96** | **1.69 *** |
| | | 95% CI | | (0.53–0.93) | (0.60–1.03) | (0.62–1.02) | (0.75–1.25) | (1.30–2.23) |
| | 2019 | cases/100,000 | 863.5 | 938.9 | 805.4 | 753.8 | 905.8 | 1132.0 |
| | | IRR | ref. | **1.09** | **0.93** | **0.87** | **1.05** | **1.31** |
| | | 95% CI | | (0.84–1.43) | (0.72–1.22) | (0.69–1.13) | (0.82–1.37) | (1.00–1.74) |
| **Risk of hospitalization from respiratory causes (J00–J99)** | 2013 | cases/100,000 | 325.0 | 322.0 | 384.7 | 594.4 | 587.9 | 694.4 |
| | | IRR | ref. | **1.00** | **1.19** | **1.85 *** | **1.83 *** | **2.15 *** |
| | | 95% CI | | (0.66–1.59) | (0.79–1.88) | (1.25–2.86) | (1.23–2.85) | (1.41–3.41) |
| | 2014 | cases/100,000 | 271.5 | 333.4 | 312.7 | 599.6 | 495.8 | 659.8 |
| | | IRR | ref. | **1.23** | **1.16** | **2.22 *** | **1.83 *** | **2.43 *** |
| | | 95% CI | | (0.78–2.04) | (0.74–1.91) | (1.45–3.59) | (1.19–2.99) | (1.54–4.03) |
| | 2015 | cases/100,000 | 327.8 | 321.6 | 310.3 | 567.4 | 481.5 | 623.0 |
| | | IRR | ref. | **1.00** | **0.96** | **1.76 *** | **1.50 *** | **1.94 *** |
| | | 95% IS CI | | (0.65–1.58) | (0.64–1.52) | (1.19–2.73) | (1.00–2.33) | (1.27–3.08) |
| | 2016 | cases/100,000 | 362.0 | 301.6 | 337.8 | 578.6 | 509.8 | 583.0 |
| | | IRR | ref. | **0.84** | **0.94** | **1.60 *** | **1.41** | **1.61 *** |
| | | 95% CI | | (0.56–1.29) | (0.63–1.44) | (1.11–2.42) | (0.97–2.15) | (1.07–2.51) |
| | 2017 | cases/100,000 | 263.8 | 372.3 | 403.2 | 512.3 | 437.2 | 647.6 |
| | | IRR | ref. | **1.45** | **1.57** | **1.99 *** | **1.70 *** | **2.51 *** |
| | | 95% CI | | (0.91–2.42) | (1.00–2.60) | (1.28–3.26) | (1.09–2.81) | (1.58–4.22) |
| | 2018 | cases/100,000 | 365.5 | 375.1 | 321.5 | 477.6 | 394.0 | 489.2 |
| | | IRR | ref. | **1.04** | **0.89** | **1.32** | **1.09** | **1.36** |
| | | 95% CI | | (0.70–1.60) | (0.60–1.37) | (0.91–2.00) | (0.74–1.67) | (0.90–2.13) |
| | 2019 | cases/100,000 | 307.5 | 298.0 | 362.0 | 623.2 | 432.1 | 515.4 |
| | | IRR | ref. | **0.96** | **1.17** | **2.02** | **1.40** | **1.67 *** |
| | | 95% CI | | (0.63–1.54) | (0.77–1.86) | (1.35–3.15) | (0.93–2.21) | (1.08–2.70) |

+ assignment into concentration intervals based on the concentrations of PM$_{2.5}$ in the year 2013, IRR—Incidence Rate Ratio, ref.—reference category for the IRR calculation, * < 0.05, CI—confidence interval.

The trend was also similar for respiratory hospitalizations; there, however, the statistically significant risk was already detected in the class with 34–35 µg·m$^{-3}$ (Table 2). However, it can be seen that the risk of cardiovascular hospitalization has been decreasing over the years. In 2019, no statistically significant difference in risk of cardiovascular hospital admissions was found even for the most polluted districts. The decreasing trend of PM$_{2.5}$ concentrations over time was also visible in the map outputs (Figures 2 and 3). Similar associations were also confirmed for respiratory causes (Table 2). In 2018 and 2019, the risk was not statistically significant (except for the ≥38 µg·m$^{-3}$ category in 2019) in any concentration interval compared to the reference interval.

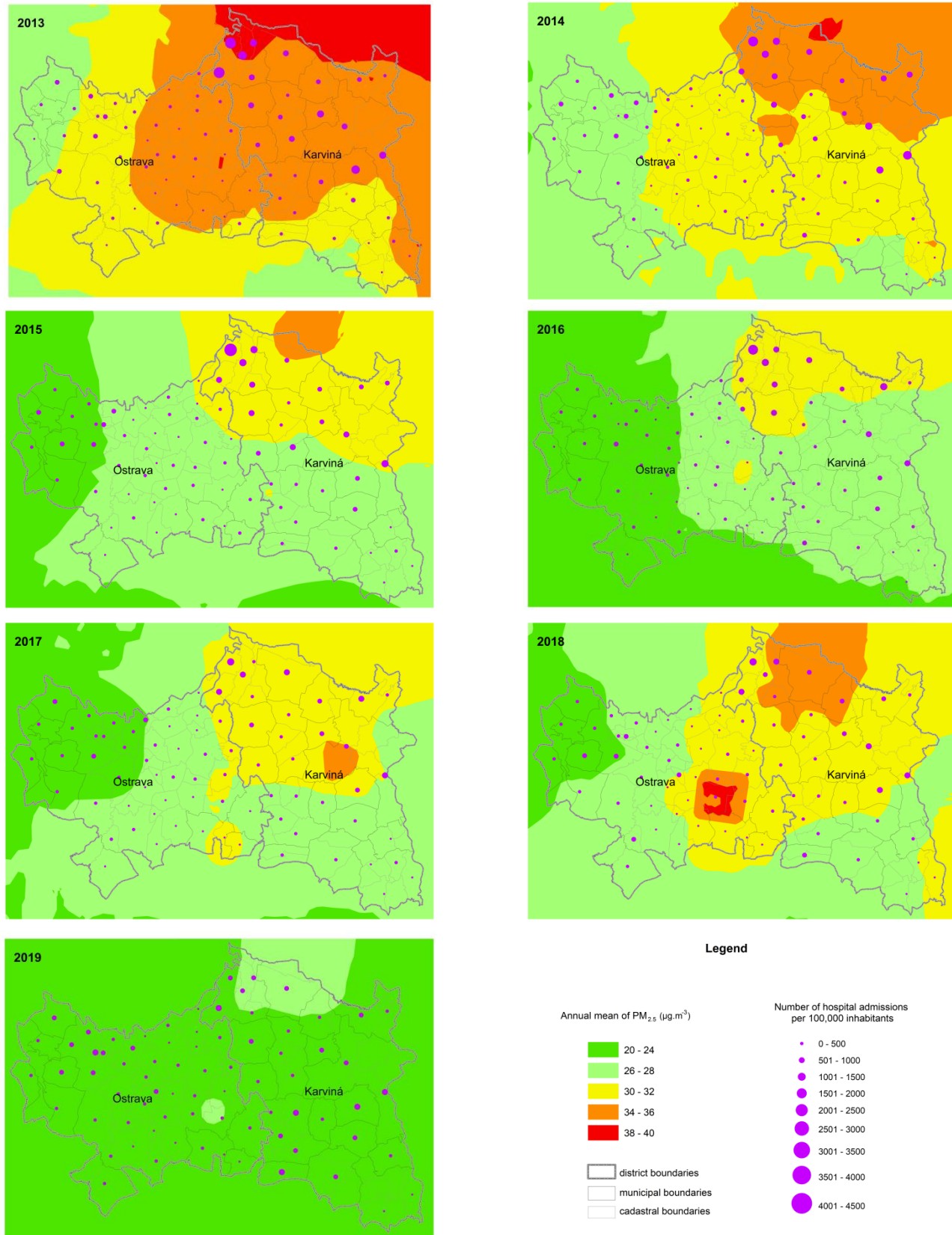

**Figure 2.** Hospital admissions for circulatory system diseases (I00–I99) per 100,000 population during the individual years of the study period.

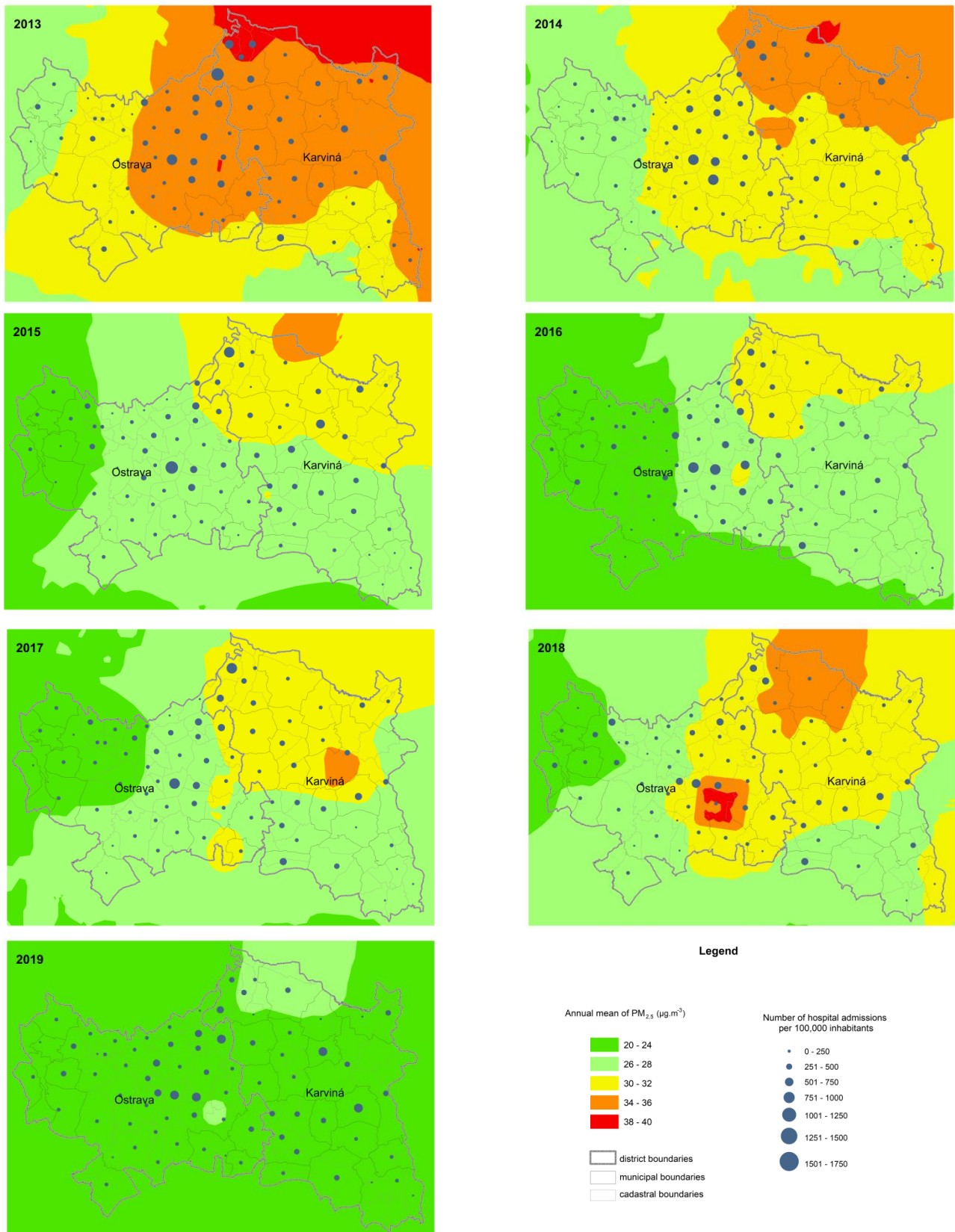

**Figure 3.** Hospital admissions for respiratory diseases (J00–J99) per 100,000 population during individual years of the study period.

## 4. Discussion

In this time-series study, we quantitatively investigated the relationships between long-term air pollution exposure and hospitalizations for cardiovascular and respiratory diseases in a heavily polluted region of the Czech Republic. Our results showed a significant $PM_{2.5}$-associated risk of cardiovascular and respiratory hospital admissions. The findings are consistent with a previous meta-analysis and individual time-series studies that reported an increased risk of hospital admissions for cardiovascular and respiratory diseases in polluted regions [5,19,20]. The other study from the same region based on the measurement data [17] found a statistically significant 3.3% increase of hospital admissions for respiratory causes per 10 $\mu g \cdot m^{-3}$ increase of $PM_{2.5}$ up to 60 $\mu g \cdot m^{-3}$ for lag 1–3. However, the fully adjusted model did not confirm statistically significant increase of hospital admissions for cardiovascular diseases associated with a 10 $\mu g \cdot m^{-3}$ increase of $PM_{2.5}$.

When identifying population exposure, the spatial homogeneity and heterogeneity of ambient air pollutants [21], and also their temporal variations [22] is a serious concern. The Czech Hydrometeorological Institute shape-file data used the concentrations of ambient air pollutants measured at individual measuring stations as the fundamental source of data for the creation of maps. The number of measuring stations is limited. In addition to the measured data, the mapping procedure also uses various supplementary data, providing complex information about the whole territory and at the same time showing the regression relation with the measured data. The main secondary source of information is represented by the models of chemical transport and dispersion of pollutants, based on the data from emission inventories and on meteorological data [18]. The GIS spatial modelling used in the presented study can, therefore, be used as a good method for concentrations assignment to cases in ecological studies.

There are numerous plausible biomedical explanations for the association between short-term exposure to fine particles and cardiovascular outcomes. The most pronounced mechanism of the adverse effect of the fine particles has been described through systemic inflammation. Particulate matter might directly or via the increased release of cytokines from alveolar macrophages trigger systemic inflammation, which is characterized by its pro-inflammatory, pro-atherogenic and pro-thrombotic changes resulting in endothelial dysfunction and atherosclerosis [23–26]. Short-term exposures to fine particles create the conditions for major deterioration of these pre-existing health issues [6]. Additionally, oxidative stress contributes to morphological changes in the lung tissue and reduced lung function. These changes may be associated with exacerbation of asthma and chronic obstructive pulmonary disease [27,28].

Our study extends the literature evidence for the impact of exposure to $PM_{2.5}$ and hospital admissions for cardiovascular and respiratory causes. It should be pointed out that most of the previous studies reported the effects of $PM_{2.5}$ as an increased risk of outcome per 10 $\mu g \cdot m^{-3}$, considering the linear concentration–response relationship between the exposure and the health outcome [5,20]. In contrast, our study presents the increased risks within five stratified exposure categories. This approach enabled us to determine the threshold concentration that represents a significant increase in the risk of hospital admission to individuals. Our results suggested that annual $PM_{2.5}$ concentrations of >34 $\mu g \cdot m^{-3}$ and >36 $\mu g \cdot m^{-3}$ were associated with an increased risk of respiratory and cardiovascular hospital admissions, respectively.

Compared to our previous study on the impact of air pollution on hospital admissions for cardiovascular and respiratory causes conducted in this urban area [16], the current study has several strengths. We substantially extended the study area to investigate the effect of $PM_{2.5}$ concentrations within the whole region (not only Ostrava city), and we prolonged the study period to 7 years (2013–2019) to prevent the influence of between-years variations of the exposures. Next, we used data from all hospitals located in this study area that minimized the potential selection bias as all cases of hospital admissions are included in the analysis. Finally, adopting the GIS technology for analysis of the modelled annual $PM_{2.5}$ concentrations using the official shape-files provided by the Czech

Hydrometeorological Institute increased the accuracy of the individual spatial exposure estimation based on the measurement data. The GIS, as the analysis tool in combination with epidemiological methods, proved to be a useful method for better evaluation of the impacts of air pollution on human health. However, this study contains some limitations as well. First, we did not examine the effect on vulnerable subpopulations (e.g., elderly population, people with chronic diseases etc.) that might be at a higher risk of experiencing adverse health effects [20,29,30]. Therefore, the findings should not be generalized for the whole population. Second, limited by the ecological design of the study, we were unable to adjust the model for the potential confounders related to individual characteristics (e.g., age, gender, smoking) [31,32] and time activity patterns (e.g., occupation, commuting, indoor air pollution) [33]. Third, we conducted only a one-pollutant model to examine the association between $PM_{2.5}$ and hospital admissions; collinearity between pollutants could, however, occur.

Our study provides public health implications. We identified the risks of cardiovascular and respiratory hospital admissions in different concentration categories using GIS spatial analysis. These findings might be used to communicate the potential risks with vulnerable populations living in the highly polluted areas of the Moravian-Silesian region.

## 5. Conclusions

The study contributes to the current knowledge about the relationships between exposures to particulate matter and acute hospital admissions from cardiovascular and respiratory causes. With increasing concentrations, the risk of both acute cardiovascular and respiratory hospitalizations increased. However, it is also necessary to continue these analyses due to the decreasing exposure of the population to $PM_{10}$ and $PM_{2.5}$ values and to further focus on monitoring $PM_1$ values.

*Key Messages*

Message 1: A statistically significant increase in the IRR for acute cardiovascular and respiratory hospitalizations was found at $PM_{2.5}$ concentrations $\geq 34$ $\mu g \cdot m^{-3}$ compared to the reference category $\leq 29$ $\mu g \cdot m^{-3}$.

Message 2: Average annual $PM_{2.5}$ concentration decreased from 30.8 to 21.4 $\mu g \cdot m^{-3}$ during the study period, which was associated with a decrease in the risk of acute hospitalization from cardiovascular and respiratory causes.

**Author Contributions:** Conceptualization, H.T., H.Š. and P.P.; methodology, H.T., H.Š. and I.T.; validation, H.T., P.P. and J.M.; formal analysis, H.T. and P.P.; resources, H.T. and P.P.; data curation, H.T.; writing—original draft preparation, H.Š. and A.D.; writing—review and editing, H.Š. and A.D.; visualization, P.P.; supervision, A.Š. and H.T. All authors have read and agreed to the published version of the manuscript.

**Funding:** This research was funded by the Technology Agency of the Czech Republic, grant number TH03030195 and by the European Regional Development Fund under the grant of the Czech Ministry of Education, Youth and Sports, the project Healthy Aging in the Industrial Environment, CZ.02.1.01/0.0/0.0/16_019/0000798 (HAIE).

**Institutional Review Board Statement:** Ethical review and approval were waived for this study due to not using personal data, as address points were anonymized by hospitals.

**Informed Consent Statement:** Not applicable.

**Data Availability Statement:** The data supporting reported results or generated during the study have been archived at the Institute of Public Health, Ostrava.

**Acknowledgments:** The authors would like to acknowledge the management and the staff of the hospitals in the area of interest for their kind collaboration with the preparation and export of the data on hospital admissions.

**Conflicts of Interest:** The authors declare no conflict of interest. The funders had no role in the design of the study; in the collection, analyses or interpretation of data; in the writing of the manuscript; or in the decision to publish the results.

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
