# Peer review of "Association between PM2.5 Exposure and Cardiovascular and Respiratory Hospital Admissions Using Spatial GIS Analysis"

_atmosphere, doi:10.3390/atmos13111797_

Round 1
Author Response
The authors thank the reviewer for the useful comments. The manuscript was changed according his/her recommendations - point by point description:
1/ The reference of the traditional time-series study originated from the same project was added both in the text, as well as in the list of references. You are right, the presented study is a descriptive study, but the use of relatively detail modelled concentrations in 1km2 cells bring a new opportunity for the epidemiological evidence within the specified areas (reference was added on the detail description of the modelling procedures and the sources of data).
2/ The authors pointed out the missing information on confounding factors; you are completely rignt the same can be said about the time varying factors. The geographical design of the study would enable stratified analysis concerning socio-economic differences between study areas. However, this is not the case as the industrial region does not show significant SES differences between the 77 units.
3/ The inclusion criteria of the hospital admissions were as follows – the first hospital admission in a particular hospital for the specified disease (principle diagnosis and the cause of hospitalization) – either cardiovascular or respiratory according the ICD-10 classification. (i.e. if the patient was transferred to another department within the same hospital, it was not the case)
4/ English language was carefully corrected, and the British English was used in the corrected version
Other comments: The introduction, method and discussion section were corrected and completed for additional information and references. Hopefully, these explanations helped to support the link between the results and conclusions.
Reviewer 2 Report
Dear Zack,
I'd like to thank you for giving me the opportunity to review this paper. Overall, the paper has been well designed, conducted and written. Prior to publication, the authors need to address the following minor comments.
As the authors might know, the most important variable in this type of study is exposure assessment and quality of air pollution data. Consequently, it is highly recommended to report hourly data coverage for PM2.5 in detail. Also, it is suggested to report the number of stations monitored PM2.5. To report these important data and information, you can consider the following paper.
https://www.sciencedirect.com/science/article/abs/pii/S0048969719341002
It is recommended to calculate the non-parametric Mann-Kendall trend test and Sen’s slope estimator so as to show temporal trend of annual PM2.5 based on the following paper.
Author Response
The authors thank the reviewer for the useful comments. The manuscript was changed according his/her recommendations - Reply to the reviewer comments:
The introduction, method and discussion section were corrected and completed for additional information and references. Hopefully, these explanations helped to support the link between the results and conclusions.
The exposure is defined based on the modelled concentrations in 1km2 cells not on the individual measured concentrations. This approach bring a new opportunity for the epidemiological evidence within the specified areas (reference was added on the detail description of the modelling procedures and the sources of data). The concern regarding the spatio-temporal variation (the recommended quotations were added) is newly discussed in the discussion section.
Round 2
Reviewer 1 Report
First, I still think this study is not novel even the authors explained that they used a model to output the exposure data within 1km. Second, there are plenty of studies of the association between PM and cardiovascular and respiratory diseases. The authors did not even compare their results with these studies. Presenting results in stratified exposure categories is not a unique strength. Previous studies usually present linear associations together with stratified results according to the percentiles of distribution.
Author Response
The authors thank to the reviewer for valuable comments. With a great respect to these comments authors tried to improve the manuscript and enclosed the revised version of the manuscript.
Ad1/ The authors agree that the used method is not novel, it has been used in the studies of the impact of polluted air on human health since the GIS system was involved in the field of epidemiology. The novelty of the presented study (and its evidence importance for one of the most polluted areas in Europe) is based on the more precise exposure data use (comparing to 8 available measurement sites of PM2.5 within the whole region) and the use of the individual data on hospital admissions assigned according to the place residence to this exposure data.
Ad2/ You are completely right regarding the plenty of studies dealing with the association between PM and CDV and respiratory diseases. The presenting evidence cannot be directly compared with these studies outputs as the analysis is different. The authors added (at the end of the first paragraph of discussion) the comparison with the results of such a type of study (investigating linear associations) based on the data from the same region.
"The other study from the same region based on the measurement data [17] found statistically significant 3.3 % increase of hospital admissions for respiratory causes per 10 µg.m-3 increase of PM2.5 up to 60 µg.m-3 for lag 1-3. However, the fully adjusted model did not confirm statistically significant increase of hospital admissions for cardiovascular diseases associated with 10 µg.m-3 increase of PM2.5."